# Artificial Intelligence: A Blessing or a Threat for Language Service Providers in Portugal

Célia Tavares , Luciana Oliveira * , Pedro Duarte and Manuel Moreira da Silva

CEOS.PP ISCAP Polytechnic of Porto, 4465-004 Porto, Portugal; celiat@iscap.ipp.pt (C.T.); pduarte@iscap.ipp.pt (P.D.); mdasilva@iscap.ipp.pt (M.M.d.S.)
* Correspondence: lgo@eu.ipp.pt

**Abstract:** According to a recent study by OpenAI, Open Research, and the University of Pennsylvania, large language models (LLMs) based on artificial intelligence (AI), such as generative pretrained transformers (GPTs), may have potential implications for the job market, specifically regarding occupations that demand writing or programming skills. This research points out that interpreters and translators are one of the main occupations with greater exposure to AI in the US job market (76.5%), in a trend that is expected to affect other regions of the globe. This article, following a mixed-methods survey-based research approach, provides insights into the awareness and knowledge about AI among Portuguese language service providers (LSPs), specifically regarding neural machine translation (NMT) and large language models (LLM), their actual use and usefulness, as well as their potential influence on work performance and the labour market. The results show that most professionals are unable to identify whether AI and/or automation technologies support the tools that are most used in the profession. The usefulness of AI is essentially low to moderate and the professionals who are less familiar with it and less knowledgeable also demonstrate a lack of trust in it. Two thirds of the sample estimate negative or very negative effects of AI in their profession, expressing the devaluation and replacement of experts, the reduction of income, and the reconfiguration of the career of translator to mere post-editors as major concerns.

**Keywords:** artificial intelligence; neural machine translation; large language models; language service providers; translators; interpreters

## 1. Introduction

Numerous parts of our personal and professional lives have undergone radical changes as a result of the ongoing development of technology. Technology has potentiated advancements in communication, enabling people to connect and interact with each other across vast distances, has democratized access to information, has transformed education, making learning more interactive, engaging, and accessible, among many other aspects. It has also given rise to new professions and industries that did not exist before. Jobs such as data scientists, app developers, cybersecurity experts, social media managers, and artificial intelligence (AI) engineers are some of the new professions that have emerged, changing the nature of work, and opening up new career prospects.

AI, for example, has changed the way individuals communicate, access, and use information, as well as how they conduct their daily lives. Notable examples include the infamous Alexa, Siri, ChatGPT, Google Translate, among many others. To corroborate AI's fast development, the McKinsey Global Survey on Artificial Intelligence (AI) [1], conducted in 2022, demonstrated that AI's growth has been bigger in the more recent years and AI adoption has, in fact, more than doubled since 2017.

AI is being used in language related areas to improve speech recognition, natural language processing, and machine translation (e.g., NMT—neural machine translation), just to mention a few examples, trying to make communication more efficient, knowledge

more widely available, and breaking down language barriers between people from different cultures and backgrounds. In other sectors of the industry, AI is being used to perform a wide variety of tasks, from simple repetitive tasks to complex decision-making processes, to increase efficiency and productivity, reduce costs, and free up human workers to focus on more creative and complex tasks.

However, AI advancements have not only provided advantages, posing some challenges and disruptions to certain professions. According to a recent study by OpenAI, Open Research, and the University of Pennsylvania, large language models (LLMs) based on AI, such as Generative Pretrained Transformers (GPTs), may have potential implications on the job market, specifically regarding occupations that demand writing or programming skills [2]. That same study identified interpreting and translation as one of the main occupations with greater exposure to AI in the US job market (76.5%). Furthermore, the already mentioned Mckinsey Survey Report also showed that activities related to "Natural-language text understanding" were third when related to functions where AI is reported to have a significant influence [1]. This may happen because AI-powered language translation tools are becoming increasingly sophisticated and accurate, and AI-powered writing tools and chatbots, such as GPTs, are also becoming more prevalent. This leads to concerns about the potential impact they may have on the demand for human translators, writers, and other LSPs in a world where language services are greatly increasing.

Given this context, this study aims to provide insight into the current perceptions of LSPs in Portugal, such as translators and interpreters, regarding AI and how it may affect their careers in the future.

## 2. Theoretical Background

### 2.1. Language Service Providers (LSPs)

The demand for language services is enormous and growing on a global scale. By the end of 2032, it is anticipated that this market, currently valued at USD60.68 billion, would have grown to USD96.21 billion [3]. Naturally, the number of language service providers (LSPs) is also huge. It encompasses organizations or individuals that offer language-related services to assist their clients in overcoming language barriers and facilitating effective communication. Some types of LSPs include translation agencies, freelance translators, interpreters, localizers, subtitlers, among many others.

Translation agencies are specialized in written language services. They employ professional translators who convert written content from one language to another while maintaining accuracy and cultural nuances. Their tasks can include a wide variety of texts from official documents to websites, marketing materials, legal texts, technical manuals, and more. Interpreters focus on spoken language services and are the ones who facilitate communication between parties speaking different languages. Interpretation services can be delivered in various settings, such as conferences, meetings, legal proceedings, healthcare appointments, among others, ensuring seamless oral communication between speakers of different languages. Localizers adapt content to suit specific target markets and cultural contexts, and localization tasks entail adapting websites, software, mobile apps, and multimedia content to make them culturally appropriate and user-friendly for different markets. Subtitlers are specialised in adding written text to audiovisual content (movies, TV shows, videos, etc.), involving transcribing spoken dialogue, translating, and synchronizing the text with the audiovisual content.

Bearing in mind the demanding context in which people live nowadays, it is important for LSPs to be technologically adapted to the current market trends. A fast online search for "Language Service Providers" indicates that numerous businesses or individuals are rising to the challenge of technological evolution and now provide a variety of services, besides the more typical ones mentioned above. Nowadays, for example, having varied degrees of MT post-editing services is quite common showing adaptability to todays' market [4]. This adaptability is paramount since, for example, the MT market is expected to reach a compound annual growth rate (CAGR) of 7.1% during the period 2022–2027 [5].

Other resources like ChatGPT are also expected to continue to grow, since they can be seen as useful resources, being capable of clarifying doubts or even explaining cultural references for translators or interpreters.

*2.2. Neural Machine Translation (NMT)*

According to Xueting and Chengze, "Artificial intelligence (AI) is a big field of studies that enables computers and machines to mimic the perception, learning, problem-solving, and decision-making capabilities of humans" [6] (p. 280). Thus, finance, healthcare, education, social services, transportation, and many other industries have already benefited from the development and advancement of AI through the use of a variety of applications, including natural language processing, translation, chatbots, text recognition, speech recognition, face recognition, image processing, and so on [6].

NMT is an artificial intelligence-based technology that uses deep neural networks [7] to translate text from one language to another automatically. It emerged in 2013 "from the encoder–decoder deep learning network structure where the encoder was a convolutional neural network, and the decoder was a recurrent neural network (RNN)" [6] (p.285). NMT models are trained on large amounts of parallel text and learn to generate translations by learning patterns of each word and sentence and identifying the most likely translation based on that context. Its most remarkable feature is the ability to actually learn, in an "end-to-end" model, the mapping of the source text associated with the output text [8].

In NMT, researchers have introduced attention mechanisms, which allow the model to focus on relevant parts of the source sentence during translation. Transformer models, introduced in 2017 [9], have gained prominence for their ability to parallelize computations and handle long-range dependencies effectively. Transfer learning techniques, such as those pretraining to large monolingual corpora [10], have also been employed to improve NMT performance. Additionally, reinforcement learning and adversarial training have been explored to enhance the quality of NMT outputs. This enables NMT systems to tackle more complex issues like syntax and grammar, which have been shown to produce more fluent and natural-sounding translations than other previous models like statistical machine translation (SMT).

NMT is widely adopted in the industry and is deployed in production systems by well-known companies like Google, Microsoft, Facebook, Amazon, RWS, Yandex, and many more [11].

*2.3. Large Language Models (LLMs)*

Large language models are one of AI's most significant developments and are trained on massive amounts of text data being able to generate coherent and relevant responses to a wide range of questions [12]. While LLMs tend to be associated with various iterations of GPT, "LLMs can be trained using a range of architectures and are not limited to transformer-based models" [2] (p. 1).

ChatGPT, from OpenAi, is based on the GPT model and is a clear example of an LLM. It is designed to generate human-like responses to natural language input, and it is often used for conversational applications, such as chatbots and virtual assistants, as well as for natural language processing tasks such as text classification, translation, and summarisation. It can also be used to generate creative writing prompts, generate new text based on existing text, and answer trivia questions.

Thus, ChatGPT's role in translation is quite relevant, since it has been trained on a vast number of texts in various languages and is able to translate text from one language to another. In this vein, LSPs may use ChatGPT for several reasons, namely for speed and efficiency since it can quickly generate translations, making it a valuable tool for translators who need to complete assignments within tight deadlines; cost-effectiveness, because it is able to translate simple or routine texts, allowing translators to focus on more complex or creative assignments; or even access to specialized terminology since it has been trained on a wide range of texts, including technical and specialized content. Although this is not

what made ChatGPT gain attention in the media, rather it was its ability to answer to an enormous variety of questions, interacting with the user, this is also a feature that users have currently been using [13].

However, and although ChatGPT seems to be the most famous tool in this domain, many others use natural language processing and machine learning techniques to generate human-like responses to user inputs. Some popular examples include Microsoft DialoGPT, Facebook Blender, Google Meena, or even Amazon Lex, to name a few.

Microsoft DialoGPT, developed by Microsoft, is an LLM designed for conversational interactions, which focuses on maintaining context and generating coherent responses [14]. Facebook Blender is a conversational AI model developed by Facebook, which aims to engage in more human-like and dynamic conversations by considering personal information and implementing a multi-turn dialogue strategy [15]. Google Meena is a chatbot developed by Google, designed to provide more meaningful and contextually relevant responses, aiming for more natural and engaging conversations [16]. Finally, Amazon Lex is a conversational AI service by Amazon Web Services (AWS), which allows developers to build chatbots and conversational interfaces with the help of natural language understanding and generation capabilities [17].

## 3. Related Research

The impact of AI in translation or other language-related professions is yet to be extensively analysed in scientific publishing. However, there are a few studies that can be mentioned.

As an example, it is possible to highlight a study conducted among Bulgarian translators to find out how concerned they were about their future and also to analyse their perceptions about the impact of AI in their profession [18]. The findings of this study made it abundantly clear that the alarmist scenarios of widespread job extinction are unjustified for Bulgaria or, at the very least, the translators do not think that their industry will be destroyed by the technological change brought on by AI. They are unconcerned because AI and other digital technology will become "organisers of new activities within the profession" [18], rather than posing a threat to already existing jobs.

Another study from 2020 analysed the effects of AI and automation on the work of communication professionals in order to determine what knowledge and expertise are required to deal with its effects [19]. The study's results indicated that, on the one hand, increased productivity and efficiency are likely to result, allowing communication specialists to concentrate on the creative side of their work as well as on strategy and analytical thinking; however, on the other hand, repetitive and low-level jobs may be lost as higher position jobs or those requiring creativity and decision-making may be more difficult to automate [19].

There is also a study conducted among 1000 professionals from different fields that shows that 63% of them believe that they would be better off thanks to artificial intelligence, regardless of their area of expertise [20].

The work conducted by Jonathan Downie [21] is also worth noting due to his reflection about the widespread concerns regarding the interpreter's professional status in a AI future.

Hence, the main purpose of this study is to analyse language service providers' perceptions regarding AI tools and their impact on LSPs' careers in Portugal.

## 4. Methods and Procedures

To achieve the proposed purpose, a mixed-methods self-administered survey-based research approach was adopted. A survey, conducted via Limesurvey, was made available in May 2023 through professional groups on Facebook like "Tradutores com Vida", translator's associations like "APTRAD—Associação Portuguesa de Tradutores" (Portuguese Translators Association), "APET—Associação Portuguesa de Empresas de Tradução" (Portuguese Association of Translation Agencies) and direct emails to institutional databases of

translators (some of which are alumni of the Polytechnic of Porto). Respondents were also encouraged to disseminate the survey to their professional networks in this area.

The questionnaire is composed of 20 questions (see Supplementary Material, File S1). However, depending on the answers, not all the questions were shown to the respondents since conditional branching was implemented. The professionals who participated in the research were contacted either directly by the researchers or through their affiliation with one of the aforementioned associations, thus they consist of a convenience sample. In Portugal there is no information available on the total population of LSPs because there is no formal board or council to regulate translators or interpreters. There are two main associations in the area (APTRAD and APT) but the number of their members is also not disclosed, which also prevented us from a making an overall estimate. However, it is also noteworthy that, as professionals, translators and interpreters can work in different legal frameworks in Portugal. They can be employed, self-employed, or even register as a company according to their labour needs. In addition, as self-employed people, translators, and interpreters can register with the tax authority using the generic designation of "service provider" and it is not compulsory to register as a translator in order to provide these services. Therefore, and considering the facts described above, it is not possible to obtain reliable data on the actual number of people carrying out this type of activity in Portugal. A total of 79 qualified responses (fully answered and submitted surveys) were gathered.

The questionnaire was anonymized in an attempt to reduce social desirability bias [22], which occurs when respondents provide answers that they believe are more socially acceptable or desirable than their actual beliefs or behaviours.

The questions in the questionnaire were categorized into three theoretical dimensions (Table 1):

**Table 1.** Definitions of the dimensions of the questionnaire.

| Dimensions | Description |
| --- | --- |
| Awareness and knowledge about AI | The degree to which a person believes he/she knows about how AI is used in their field of work (awareness) and effective knowledge demonstrated when tested (knowledge) |
| Actual use and usefulness | Effective use/incorporation of AI in job-related tasks (use) and perceived value-added of AI (usefulness) |
| Influence on work performance and labour market | Perception of the effects of AI on daily work (work performance) and perceived effects on the future of the labour market, global and local |

Quantitative data were extracted from LimeSurvey and imported into IBM SPSS Statistics 28, where they were pre-processed and subjected to descriptive statistics and parametric statistical tests, namely the Student *t*-test for two independent samples and one-way ANOVA for tests with more than two samples. A significance level of less than 0.05 was considered for the tests. Qualitative data were imported to MaxQDA, 2020, and submitted to inductive content analysis.

## 5. Results and Discussion

### 5.1. Sample Demographics

The sample demographics are depicted in Table 2. Most of the professionals are female (75.9%), with a degree (50.6%), living in Portugal, aged over 41 years old, working as freelance translators, mostly full-time, or as interpreters, and with more than ten years of experience in the career.

**Table 2.** Sample demographics.

| Gender | n | % |
|---|---|---|
| Male | 17 | 21.5 |
| Female | 60 | 75.9 |
| Non-binary | 3 | 2.5 |
| **age** | | |
| <26 | 7 | 8.9 |
| 26–30 | 5 | 6.3 |
| 31–35 | 3 | 3.8 |
| 36–40 | 11 | 13.9 |
| 41–45 | 17 | 21.5 |
| 46–50 | 18 | 22.8 |
| >50 | 18 | 22.8 |
| **country** | | |
| Portugal | 76 | 96.2 |
| Spain | 1 | 1.3 |
| Other [a] | 2 | 2.5 |
| **education** | | |
| Non-graduate | 3 | 3.8 |
| Degree | 40 | 51.3 |
| Master or PhD | 35 | 44.9 |
| No answer | 1 | 1.3 |
| **job titles** | | |
| Freelance translator (full-time) | 39 | 49.4 |
| Freelance translator (part-time) | 24 | 30.4 |
| In house translator | 8 | 10.1 |
| Translation project manager | 5 | 6.3 |
| Owner of a translation agency and/or language services | 9 | 11.4 |
| Interpreter | 19 | 24.1 |
| Proof-reader/Post-editor | 13 | 16.5 |
| Subtitler | 11 | 13.9 |
| Localizer | 3 | 3.8 |
| Other [b] | 3 | 3.8 |
| **years of experience** | | |
| <5 | 10 | 12.7 |
| 5–9 | 9 | 11.4 |
| 10–14 | 12 | 15.2 |
| 15–19 | 12 | 15.2 |
| 20–24 | 14 | 17.7 |
| 25–29 | 11 | 13.9 |
| >29 | 11 | 13.9 |

Notes. [a] Portuguese, living in Italy. [b] Also a teacher.

The professionals were asked to mark all their job titles (multiple choice), as it is common in the career to accumulate more than one title. As a result, fifteen unique job title combinations were found for the professionals with more than one job title, as depicted in Table 3. The job title "Translator" is persistent in all the combinations and most frequently combined with "Interpreter" (16.46%), "Subtitler" (6.33%) and "Proofreader/Post-editor" (5.06%).

**Table 3.** Unique combinations of job titles in the career.

| Job Titles | n | % |
|---|---|---|
| Translator, Interpreter | 13 | 16.46 |
| Translator, Subtitler | 5 | 6.33 |
| Translator, Proof-reader/Post-editor | 4 | 5.06 |
| Translator, Proof-reader/Post-editor, Subtitler | 2 | 2.53 |
| Translator, Owner of a translation and/or language service agency | 2 | 2.53 |
| Translator, Proof-reader/Post-editor, Localizer | 1 | 1.27 |
| Translator, Subtitler, Localizer | 1 | 1.27 |
| Translator, Translation project manager, Owner of a translation and/or language service agency, Proof-reader/Post-editor, Subtitler | 1 | 1.27 |
| Owner of a translation and/or language service agency, Proof-reader/Post-editor, Subtitler | 1 | 1.27 |
| Translator, Interpreter, Proof-reader/Post-editor | 1 | 1.27 |
| Translator, Translation project manager, Owner of a translation and/or language agency, Proof-reader/Post-editor Freelance Translator, Localizer | 1 | 1.27 |
| Translator, Owner of a translation and/or language service agency, Interpreter | 1 | 1.27 |
| Translator, Freelance Translator, Interpreter | 1 | 1.27 |
| Translation project manager, Owner of a translation and/or language service agency, Interpreter, Proof-reader/Post-editor, Subtitler | 1 | 1.27 |

### 5.2. Awareness and Knowledge about Artificial Intelligence

The professionals were asked about their level of familiarity with artificial intelligence in performing language services (translation, interpreting, etc.) and were tested regarding their knowledge of AI-powered software and applications.

As observed in Figure 1, approximately 67% of the professionals are moderate to very familiar with using AI in performing language services, with only a small portion of 10% not being familiar at all. These 10% might mean that the respondents are unaware of AI prevalence in tools like Google Translator or DeepL since professional translators would hardly have never used them. A series of $t$-tests and one-way ANOVA tests showed no significant differences in the average of familiarity per gender, age group, and work experience. It is worth noticing, however, that the greater portion of the professionals are only moderately familiar with AI in performing their services, which might mean that they are still reluctant to use machine translation tools.

For the professionals who are at least slightly familiar with the use of AI in performing language services ($n = 71$), we tested (quiz-like questions) how they would classify the most common commercial and free translation tools in two categories—AI and automation—to

verify their level of knowledge about the relationship of the tools to the technologies. The frequencies of votes per tool are presented in Table 4.

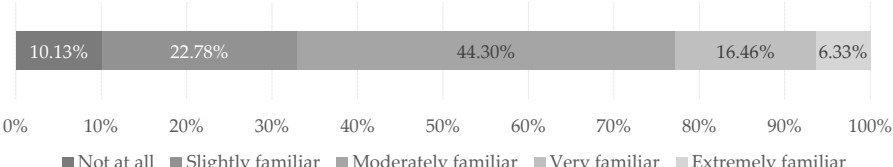

**Figure 1.** Level of familiarity with the use of AI in performing language services.

**Table 4.** Frequencies of votes per tool (%).

|  | AI | Automation | Both | None | Don't Know |
|---|---|---|---|---|---|
| ChatGPT | 63.3 | 0 | 22.8 | 1.3 | 2.5 |
| DeepL | 17.7 | 32.9 | 30.4 | 0 | 8.9 |
| Google Translate | 11.4 | 40.5 | 27.8 | 0 | 10.1 |
| Microsoft Translator | 7.6 | 29.1 | 24.1 | 2.5 | 26.6 |
| SmartCat | 5.1 | 31.6 | 16.5 | 3.8 | 32.9 |
| Trados Studio | 1.3 | 49.4 | 17.7 | 6.3 | 15.2 |
| MemoQ | 3.8 | 46.8 | 16.5 | 5.1 | 17.7 |
| Phase | 3.8 | 16.5 | 12.7 | 2.5 | 54.4 |
| Wordfast | 2.5 | 36.7 | 8.9 | 5.1 | 36.7 |

All of the tools provided in the list use both technologies—AI and automation—thus we scored the answers as follows: "AI" = 0 points, "Automation" = 0 points, "Both" = 1 point", "None" = 0 points, "Don't know" was coded as missing value. The total score per person consists of the sum of the points obtained in each of the tools mentioned in Table 4, varying from 0 to 9. The frequencies of unique scores obtained by the professionals are presented in Table 5. The global average score is $\bar{x} = 1.97$ (Min.: 0; Max.: 9 points).

**Table 5.** Frequencies of unique scores and grades (*n* = 71).

| Scores [a] | N | % | Assigned Grade | Grade % |
|---|---|---|---|---|
| 0 | 26 | 36.6 | Very Poor | 54.9 |
| 1 | 13 | 18.3 | | |
| 2 | 10 | 14.1 | Poor | 18.3 |
| 3 | 3 | 4.2 | | |
| 4 | 8 | 11.3 | Fair | 19.7 |
| 5 | 6 | 8.5 | | |
| 6 | 2 | 2.8 | Good | 4.2 |
| 7 | 1 | 1.4 | | |
| 8 | 1 | 1.4 | Excellent | 2.8 |
| 9 | 1 | 1.4 | | |

Note. [a] Maximum score = 9 (nine tools).

As observed, the global average is very low ($\bar{x}$ = 1.97 points out of 9), 54.9% of the professionals scored between zero and one point (Very Poor) and 18.3% scored between two and three points (Poor). In our sample, 73.2% of the professionals were unable to identify whether AI and/or automation technologies support the tools that are most used in the profession. We should note that we did not evaluate the level of proficiency for each

tool. However, we provided a short list of the most common commercial and free tools (competitors) that we believe are mostly known in the field. These results are, somewhat, in line with [18].

### 5.3. Actual Use and Usefulness

The professionals were asked about the frequency of use of two of the most relevant AI technologies used on the job—NMT and LLM—and their motivations to use them. Regarding the frequency of use (Table 6), NMT $\bar{x} = 3.32$ is almost two times more frequent than LLM $\bar{x} = 1.86$. This may be explained by how LLM have recently become popular (e.g., ChatGPT in 2023), while NMT has been available for longer.

**Table 6.** Frequency of use of AI technologies on the job.

|  | N | Min. | Max. | $\bar{x}$ | σ |
|---|---|---|---|---|---|
| Neural Machine Translation | 79 | 1 | 6 | 3.32 | 1.364 |
| Large Language Models | 79 | 1 | 6 | 1.86 | 1.195 |
| Artificial Intelligence [a] | 79 | 1 | 6 | 2.59 | 1.091 |

Notes. [a] Index. Scale: 1 = Never—0%; 2 = Rarely—Up to 25%; 3 = Sometimes—Between 26% and 50%; 4 = Often—Between 51% and 75%; 5 = Very Often—Between 76% and 99%; 6 = Always—100%.

The overall frequency of use of artificial intelligence technologies falls under the "Sometimes" interval, with professionals using them about 25% to 50% of the time.

A one-way ANOVA test confirms that men $\bar{x} = 2.00$ use language models more frequently than women $\bar{x} = 1.72$ ($F(10.643) = 8.969$; $p < 0.0001$). A series of one-way ANOVA tests also confirms that the frequency of use of NMT, language models and AI technologies overall is higher for professionals whose level of familiarity is also higher ("Very familiar" and "Extremely familiar"). No significant differences were found between age groups or years of experience on the job.

When asked about the reasons why professionals use NMT, the top five reasons reside in saving time ("Speed", 55.7%), serving as a starting point for the translation of texts (35.4%), which allows professionals to focus on post-editing tasks (25.3%). Being useful for short texts (26.6%) and allowing generic understanding of the text is also referred to (22.8%), as depicted in Table 7.

**Table 7.** Usefulness of NMT (%).

| Usefulness of NMT | *n* | % |
|---|---|---|
| Speed | 44 | 55.7 |
| Works as a starting point for my translation | 28 | 35.4 |
| It is useful for short text excerpts | 21 | 26.6 |
| I prefer to focus my efforts on post-editing tasks (proofreading for terms, grammar, style, etc.) | 20 | 25.3 |
| Allows me to understand the text in a generic way | 18 | 22.8 |
| It is useful for more difficult passages of text | 16 | 20.3 |
| Provides translation results appropriate to the message | 13 | 16.5 |
| It is useful for texts with few cultural/contextual aspects | 12 | 15.2 |
| It is useful when I need to translate into a foreign language | 9 | 11.4 |
| It is useful when I need to translate into my mother tongue | 4 | 5.1 |
| Provides a better translation than I would | 2 | 2.5 |
| **Other** | | |
| To provide a broader range of synonyms/alternative words | 2 | 2.5 |
| Company requisite | 1 | 1.3 |
| To reduce production costs for the client | 1 | 1.3 |

On top of the suggested list of potential motivations, professionals also refer to the larger variety of vocabulary provided by NMT and its ability to save costs for the client. There is one case in which it consists of a company requisite.

Concerning the usefulness of LLM, the reasons for their use are in line with the general motivation to use NMT (Table 8). The most common utility resides in saving time in research (16.5%) and providing more translation options (12.7%). On top of the provided list of potential reasons, the professionals also refer to its ability to verify the clarity of the translated text in more complex sentences and to reduce costs for the client. Additionally, some, at present, are using the tool to evaluate the potential of the technology or out of curiosity.

**Table 8.** Usefulness of language models (%).

| Usefulness of Language Models | *n* | % |
|---|---|---|
| To save research time in general | 13 | 16.5 |
| Provides more translation options | 10 | 12.7 |
| To understand specific terminology | 8 | 10.1 |
| To understand cultural references | 6 | 7.6 |
| **Other** | | |
| To verify the clarity of the translated texts, in long, complex sentences | 1 | 1.3 |
| To evaluate the potential of the technology | 1 | 1.3 |
| Out of curiosity | 1 | 1.3 |
| To reduce production costs for the client | 1 | 1.3 |

The professionals were also asked if they felt the AI technologies, in general, were useful in their specific job attributions. The results, in Figure 2 show that only about 34% of professionals consider the technologies as very or extremely useful, and 15,19% find them not useful at all.

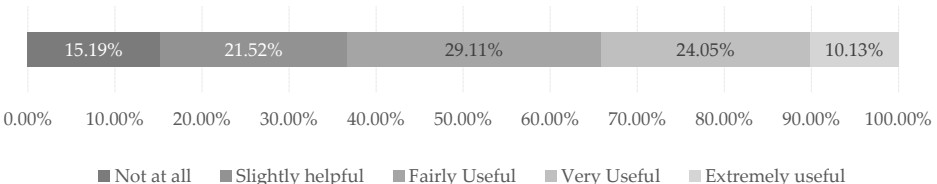

**Figure 2.** Usefulness of AI in the performance of job attributions.

A series of *t*-tests and one-way ANOVA tests reveal that there are no significant differences in the distribution of the usefulness averages among professionals with more or less years of experience. However, there are substantial differences among age groups and among the level of familiarity with AI, as shown in Figure 3.

Professionals aged <26 years and between 46 to 50 years old recognize higher levels of usefulness in AI ($F(3.922) = 3.069$; $p < 0.005$). Usefulness is also higher for the professionals who are very or extremely familiar with AI ($F(6.468) = 5337$; $p < 0.05$). As mentioned, AI technologies have become more popular recently and, in our sample, 44% of professionals are only moderately familiar with it and 33% are only slightly or not familiar with it at all. We believe that as the professionals become more familiarized with AI, they tend to see more usefulness in it. In fact, a one-way ANOVA test confirms that the participants who scored poorer in the quiz about AI and automation also reveal lower averages of perceived usefulness of AI in the performance of their work ($F(2.625) = 2.154$; $p < 0.05$).

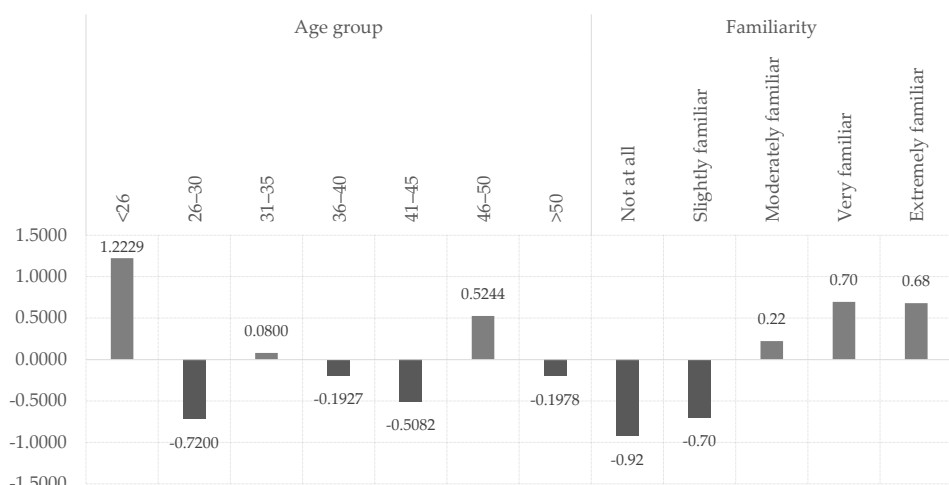

**Figure 3.** Usefulness of AI in the performance of job attributions per age group and familiarity (standardized values).

For the participants who stated that AI is, at least slightly useful (*n* = 67), the most relevant reasons among the provided list consist of "Speed" and "Efficiency" (Table 9).

**Table 9.** Reasons why AI is useful (*n* = 67).

| Usefulness of AI | *n* | % |
|---|---|---|
| Speed | 61 | 91.0 |
| Efficiency | 31 | 46.3 |
| Good value for money | 15 | 22.4 |
| **Other** | | |
| Good suggestions of synonyms and alternatives | 2 | 3.2 |
| Helps to understand certain contexts | 1 | 1.5 |
| It helps to connect ideas | 1 | 1.5 |

For the participants who stated that AI is not useful at all (*n* = 12), the most relevant reasons, among the provided list, consist of "Lack of contextual understanding" (the inability of the tool to understand the context of the texts) and "Lack of precision and quality" (Table 10).

**Table 10.** Reasons why AI is not useful (*n* = 12).

| Useless of AI | *n* | % |
|---|---|---|
| Lack of contextual understanding | 9 | 75.0 |
| Lack of precision and quality | 7 | 53.3 |
| Lack of confidentiality guarantee | 4 | 33.3 |
| **Other** | | |
| Don't know how to use it | 1 | 8.3 |

Overall, we observe that the professionals are not very familiar with the use of AI in the profession and that their knowledge regarding these technologies and tools is low to very low. As a result, their perceived level of usefulness of AI is essentially low to moderate and those who are less familiar, know less, and recognize less usefulness also demonstrate a lack of trust in AI, as observed in the reasons depicted in Table 8.

*5.4. Influence on Work Performance and the Labour Market*

To evaluate the influence of AI on their work performance, the professionals were asked what they thought was the influence of the technology on specific job tasks in the present and the language services market in the future.

Concerning the influence of AI on job-specific tasks, the results are presented in Table 11.

**Table 11.** Influence of AI on job-specific tasks.

|  | *n* | Min. | Max. | $\overline{x}$ | σ |
|---|---|---|---|---|---|
| Transcription | 45 | 1 | 4 | 2.73 | 0.809 |
| Translation of general topics | 53 | 1 | 4 | 2.72 | 0.841 |
| Terminology management | 41 | 1 | 4 | 2.56 | 0.743 |
| Technical/specific translations | 61 | 1 | 4 | 2.46 | 0.765 |
| Post-editing | 45 | 1 | 4 | 2.42 | 0.941 |
| Revision | 44 | 1 | 4 | 2.18 | 0.870 |
| Subtitling | 33 | 1 | 4 | 2.09 | 0.805 |
| Interpretation | 33 | 1 | 4 | 1.94 | 0.827 |

Notes. Scale: 1 = "Very negatively"; 2 = "Negatively"; 3 = "Both"; 4 = "Positively"; 5 = "Very positively".

The tasks the professionals believe benefit more from AI are "Transcription", "Translation of general topics", "Terminology management" and "Technical/specific translations". Those with less positive effects of AI consist of "Interpreting" and "Subtitling", probably due to the particular characteristics of these tasks. In interpretation, there are many variables that may influence the correct understanding of what is being said (tone, pronunciation, noise, etc.) and in subtitling, there are a set of rules that are also important to convey in the message, namely the number of characters and lines. It is worth noting that the overall average for all tasks is relatively moderate $\overline{x} = 2.11$, falling in the "Negatively" to "Both" interval.

After a series of *t*-tests and one-way ANOVA tests, we found no significant differences in the distribution of averages between genders, age, work experience, familiarity with AI, frequency of use of NMT and language models, and usefulness and frequency of use of AI.

Concerning the influence of AI on the language services market in the future, the professionals were provided with the following answer options:

(1) Very negatively: I believe that AI will have a highly negative impact on the language services market, which may include the devaluation of language professionals, the loss of linguistic nuance, or an erosion of the human touch in language services.

(2) Negatively: I foresee potential drawbacks, such as displacement of jobs, reduced demand for human translators and interpreters, or a decrease in the quality of language services.

(3) Both: It can bring both opportunities and challenges to the market.

(4) Positively: I foresee benefits such as increased productivity, streamlined processes, and the ability to handle larger volumes of content.

(5) Very positively: I believe it will bring significant advances, such as greater efficiency, better quality, and more opportunities for language professionals.

Only 38% of the professionals estimate a positive or very positive outcome, against a fraction of 46% who estimates a negative or very negative one, and an additional portion of 16% who estimates it can turn out either positive or negative (Figure 4).

These results are fundamentally different from those obtained in Bulgaria [18], which report that only about one tenth of respondents (12%) are of the opinion that translators would become redundant.

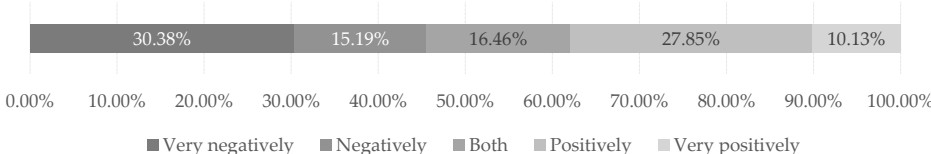

Figure 4. Influence of AI the language services market.

A series of one-way ANOVA tests (standardized values illustrated in Figure 5), allowed us to identify that this overall negative tendency is essentially voiced by the professionals with below-average familiarity with AI in the provision of language services ($F(5.791) = 7.351$; $p < 0.001$), below-average knowledge regarding the incorporation of AI and automation in language tools ($F(5.064) = 5.537$; $p < 0.001$), below-average frequency of use of AI technologies on the job ($F(6.553) = 7.274$; $p < 0.001$), below-average perceived usefulness of AI in the provision of language services ($F(12.520) = 14.153$; $p < 0.05$) and perceived negative influence of AI on job-related tasks ($F(1.942) = 11.358$; $p < 0.05$).

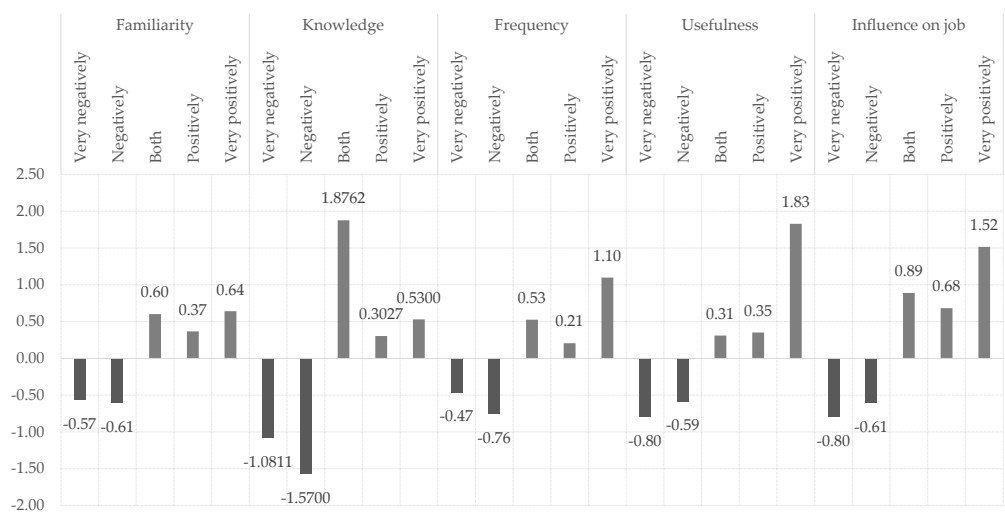

Figure 5. Influence of AI the language services market, per usefulness, familiarity, frequency of use and influence on job-related tasks (standardized values).

The professionals were also asked about their perspective concerning the future of the language services market in Portugal. They were presented with the following open-ended question: "According to a recent study [2], artificial intelligence (AI)-based tools such as ChatGPT may have implications for the language services market. This study identified interpreters and translators as some of the top professions with the greatest exposure to AI in the US labour market (76.5%), noting that AI could save professionals a significant amount of time in getting work done and/or lead to a high portion of work being automated and replaced by technology. According to your experience in the field of language services, what do you think about this issue for Portugal?".

A total of 44 documents (answers) obtained from the experts were submitted to induction-based quantitative content analysis in MaxQDA. The total number of answers were provided by experts that had chosen the following answers regarding the influence of AI in the language service market (Table 12).

There are eighteen answers on the positive side ("Positive" + "Very positive") and eighteen on the negative side ("Very negative" + "Negative"). The codes' map obtained from MaxQDA is presented in Figure 6, showing where the most frequent categories are and their relationships.

**Table 12.** Open-ended answers regarding the future influence of AI in the Portuguese language market, per type of influence.

|  | *n* | *%* |
|---|---|---|
| Positive (P) | 15 | 34.09 |
| Very negative (VN) | 12 | 27.27 |
| Both (B) | 8 | 18.18 |
| Negative (N) | 6 | 13.63 |
| Very positive (VN) | 3 | 6.82 |

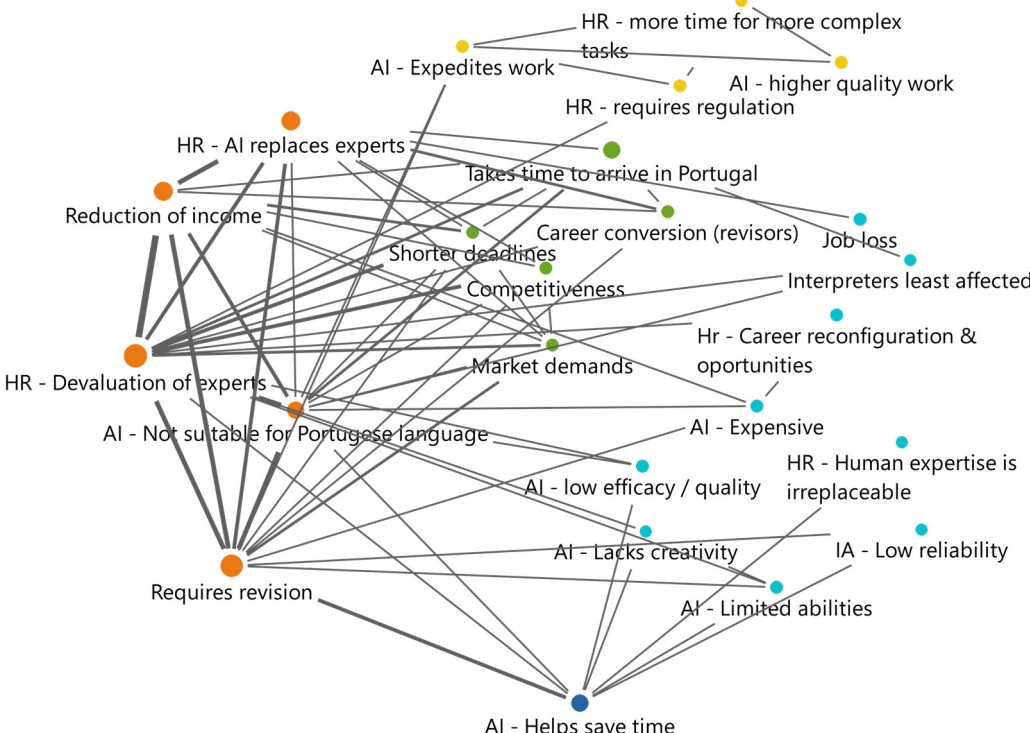

**Figure 6.** Codes' map for the future implications of AI in the Portuguese language services market.

The codes with the highest degree of saturation consist of those in the orange cluster, which refers to the most critical negative influences that experts foresee for the Portuguese language market. These are depicted in Table 13.

**Table 13.** Most critical negative outcomes of the penetration of IA in the Portuguese language market.

|  | *n* | *%* |
|---|---|---|
| Devaluation of experts | 14 | 31.82 |
| Automatic translation requires human revision | 13 | 29.55 |
| Reduction of income | 9 | 20.45 |
| AI replaces experts | 9 | 20.45 |
| Effects are not immediate; will take some time to be felt in Portugal | 7 | 15.91 |
| Not suitable for Portuguese language | 7 | 15.91 |
| Helps to save time | 7 | 15.91 |

Even though the answers were obtained from a balanced sample of experts, regarding the positive or negative orientation on the influences of AI in the langue services market,

when considering the effects on the Portuguese market, the negative outcomes appear as more relevant. This happens because, even for those experts that believe that the influences will, in general, be more positive, they also voice more general or more specific concerns with a non-positive outlook, which translates, somewhat, into a negationist orientation. The devaluation of experts and their career is at the top of these concerns and appears, sometimes but not only, to be linked to the lack of formal regulation of the career. This concern is frequently linked to other concerns, which we depict on Table 14.

**Table 14.** Quotations on concerns associated with the devaluation of experts.

| | |
|---|---|
| Lack of quality standards | *Given that language professionals are already highly undervalued in Portugal, I honestly don't see anything good on the horizon. This lack of valuation, coupled with the ability of tools like ChatGPT to present a readable and convincing text to laypeople that actually contains several errors, represents a possibility that language service providers will slowly be supplanted, because the job market values quantity over quality.* (N03, Pos. 1) |
| Lack of quality standards | *I fear that the advances in AI may diminish the importance of translators, as many customers will be satisfied with the result of machine translation.* (VN04, Pos. 1) |
| Market demands | *There should be a control and a way to value the profession (and not only the translators, for example Stable Diffusion is taking jobs away from illustrators and there are AIs that imitate actors' voices very well, there are already companies asking voice-over actors to sign a contract so they can use their voice indefinitely). There has to be respect for people who have taken so long to cultivate their career.* (VN02, Pos. 1) |
| Lack of regulation | *I have an opinion, however depending on the development of IA and regulation of our activity it can both bring opportunities and challenges to the industry. It is an uncertainty... if on the one hand it will threaten the profession, on the other hand, it can increase the volume of what is translated in the world. The big threat is the lack of regulation and the direct access of the general public who don't understand the nuances associated with these processes.* (B07, Pos. 1) |
| Lack of regulation | *There are now tools that identify whether a text is made by AI or not, which although not 100% accurate, in the future will be better. Rules now need to be implemented, and as quickly as possible. In my case, I am relatively new, but in the case of colleagues who have been in the career longer, this could have unimaginable repercussions, and again, not only to translators.* (VN02, Pos. 1) |
| Depreciation of tasks | *We will become mere text editors, earning less and there will be less work for everyone, and in some cases not even that, because the client will blindly trust AI, as some already do today with Google's translator.* (VN10, Pos. 1) |

The lack of formal regulation to sustain the relevance of the experts to the market and ensure the quality of the translated texts appears as one of the reasons why they anticipate the devaluation of the profession. Regarding this, some experts foresee a shift in

the career to become "mere text editors" (VN10, Pos. 1), which is also linked to the second most critical negative influence expected ("Automatic translation requires revision", in Table 13), as the clients tend to lower or adjust quality standards to incorporate machine translations. This appears both as a risk and as justification to sustain the role of the experts in the language services market, as is in line with [18], who reports that 88% of Bulgarian translators think that AI will not be able to perform better translations than humans, as clarified in the following quotations:

1.  *There will always have to be a human hand in the language services.* (N04, Pos. 1)
2.  *Review will always be necessary.* (N02, Pos. 1)
3.  *The intervention of the translator will always be necessary.* (VP01, Pos. 1)
4.  *There is the safeguard that for now, the tool is not perfect and often gives wrong information, so I always have to check the text, but in the future who knows.* (VN02, Pos. 1)
5.  *I'm already sensing that some agencies use these tools and send the translated texts to me for proofreading when it comes to more technical jargon. I assume that in more general text they don't even do that.* (VN06, Pos. 1)
6.  *More and more translation companies (and others) are resorting to the use of AI-based tools for translating their work, and this is something that is already implemented in Portugal. It can be useful in the sense that it provides a faster translation with some reliability, particularly in the case of technical texts, and even allows some time to be saved or dedicated to the PED and revision work, but a human translator or reviser will always be needed to correct faults and problems of interpretation and contextualization.* (P04, Pos. 1)
7.  *The chances of Artificial Intelligence (AI) negatively affecting work in Portugal are getting higher and higher. This aspect has become more recognizable, especially in translation for subtitling. In conversations with people who use AI more often, even if in a more relaxed context, and not for professional purposes, it is still learning Portuguese from Portugal, being already quite "fluent" in Brazilian Portuguese, and this is noticeable in subtitling series and movies, mostly on the Netflix platform. With the constant creation of new movies and series, the need for fast and cheap subtitles grows, so AI is used to try to "save a few bucks" with a mediocre translation, even though it is constantly being learned, leaving the rest of the work to the proofreader.* (VN03, Pos. 1)
8.  *AI is quite limited when it comes to localization or similar work. Passing a text that needed localization by AI might result in more work fixing what was wrong.* (P01, Pos. 1)

This line of reasoning is also frequently associated with the AI not being fully equipped to deal with European Portuguese (tends to be more efficient in Brazilian Portuguese), not being able to identify cultural references and being ineffective in specialized texts (e.g., literary translations). Some domains also appear to be more prone to being replaced by AI, such as translation for subtitling (according to (VN03, Pos. 1)). However, this is not unanimous because, as seen on Table 11, subtitling is one of the job tasks that, on average, is considered to be the least influenced by AI. A residual number of experts, namely those with a more positive orientation towards the future of the career, foresee that this shift will turn AI into a resource that will strengthen the career, by providing experts with tools that allow them to work faster and increase their competitiveness.

However, the transition of the role of experts to mere reviewers is one of the critical and more voiced concerns, that, together with the lowering of quality standards and increased speed of translation, will cause a severe reduction in prices. It is evident that machine translation leads to a shift in the market dynamics and in the labour market, depreciating the perceived value of language services and the value of experts, as voiced in the excerpts in Table 15. This, however, is contrary to the report of [18], according to which 59% of Bulgarian translators believe that their services will not become cheaper because of AI.

Ultimately, some experts expect AI and automation to fully replace translators, leading to the disappearance of the career, as visible in the excerpts below:

1.  *Only those who live on the moon think that the translation profession has a future.* (VN07, Pos. 1)
2.  *It will replace translators, unfortunately!* (N06, Pos. 1)

3. *You will not be able to save time, you will simply replace translators and interpreters. I already know companies in the area that have laid off hundreds of people.* (VN08, Pos. 1)
4. *AI will accelerate the decline and disappearance of our profession.* (VN10, Pos. 1)
5. *It will probably replace translators in the short/medium term, negatively influencing the translation market in Portugal.* (VN11, Pos. 1)
6. *Taking more or less time, I believe that the same will happen in Portugal. I believe that much of the work may be effectively replaced by technology, which will lead many translators to become, essentially, proofreaders. I don't think the impact will be negative, but rather challenging, as far as translators are concerned.* (B01, Pos. 1)
7. *Personally, I like to think that there will always be a handful of Portuguese companies/clients left who will have some discernment and will continue to hire human translators, because they know that translations often need that invisible human element that AI will never replace.* (B06, Pos. 1)

**Table 15.** Quotations on concerns associated the reduction in income and reconfiguration of the career.

| | |
| --- | --- |
| | *It's also going to lead to lowering fares to 1/4, which shouldn't happen.* (N02, Pos. 1) |
| | *It will lead to a lowering of fares.* (P07, Pos. 1) |
| | *As a real example, I recently received a job offer to feed words into an AI. The price was 2 cents per word. Besides the insult that the price was, you have to take into account that I would be working, to take away jobs from myself and so many others in the future. Nevertheless, there will always be someone who accepts that.* (VN02, Pos. 1) |
| Reduction in income | *The machine is already heavily relied upon, everything that leads to cost reduction takes precedence over the human factor.* (VN07, Pos. 1) |
| | *For Portugal it will probably be the same, especially since most customers of language services want a cheap service without much concern for quality. For this, AI is the ideal answer.* (B03, Pos. 1) |
| | *The mere presence of the AI in our daily lives makes the art and craft of the profession invisible, leading to proposals that are inadequate to the existing "workforce". This mismatch is certainly expressed in the salary and weekly hours to be worked.* (B04, Pos. 1) |
| Reconfiguration of the career | *I foresee a redefinition of the profession, the need for continuous training for professionals to adapt to other realities. I foresee the acceleration of processes, and I think that associated with this will come a price adjustment.* (P13, Pos. 1) |
| | *Negative impact on current professions, but probably positive on the creation of related professions.* (P10, Pos. 1) |

It is worth noting that these concerns are voiced both by experts with a more negative and uncertain foresight of the future of the language services market in Portugal. This means that, even though the potential of AI is recognized as a competitive factor for experts providing language services, namely in the expediency of the services provided and in the richness of the vocabulary, the experts fear they might end up being completely dispensable.

The study conducted on Bulgaria [18] had also identified that most of the professionals perceive artificial intelligence and automatization as threats to the profession, believing that AI will modify the profession by relieving human translators of the routine, technical part of the job, while moving translators to predominantly edit machine-translated texts, and teach artificial intelligence to perform machine translation. However, in Bulgaria, the pessimistic scenarios about mass job destruction are reported as unjustified.

The apprehension and concerns regarding the downfall of the career appear to be somewhat eased by the argument that the penetration of AI in the Portuguese language market has not reached its full potential yet and should be slower than for other countries

(English speaking, for instance), as stated by one of the experts "*It will take years to have a negative effect on our business*" (B08, Pos. 1). This refers to the incorporation of technology in the market and to AI's efficacy for dealing with the Portuguese language specifically, which experts indicate to be limited and questionable, as voiced by several other professionals:

1. *The Portuguese language is quite complex, so it will take longer to perfect the results and it will always require the intervention of a translator, but it makes the work much easier and faster.* (VP01, Pos. 1)
2. *It is more difficult to get a quality translation that captures cultural aspects, for example, into Portuguese. It is also harder to get good results for pt_PT. Since the new spelling agreement has not yet been adopted by many, there may be inconsistencies that AI cannot resolve. However, in more general texts, it is undoubtedly a help to the translator.* (P08, Pos. 1)
3. *I think that the fact that there are not yet enough corpora in Portuguese will delay this trend somewhat, but I have no doubt that people who do not understand the work and importance of a translator will prefer to use this service.* (VN04, Pos. 1)
4. *Tt ends up standardizing the content a bit, and the market may react to that, especially in areas where more creativity and distinction in the content is required.* (P07, Pos. 1)

We have, therefore, identified two delayers: cultural (adoption of AI) and technological (perceived efficacy). From the previous excerpts, we have also identified a sense of hope regarding the maintenance of the human intervention in the provision of language services or to exalt it, as depicted in the quotations below, and they ask for protective regulation, as seen in Table 14:

1. *Let's remember, being a translator and interpreter is a craft, not a merely dispensable act. Besides, humans have qualities (virtues, defects and nuances) that an AI will never be able to distinguish. This is a very personal opinion.* (B04, Pos. 1)
2. *AI could save translators and interpreters a significant amount of time in doing the work. However, it is important to note that AI does not replace the experience, practice, technique, expertise, know-how, competence, skills, capacity, knowledge of translators and interpreters.* (P14, Pos. 1)

These refer mainly to human qualities that the experts believe AI cannot offer and to technical abilities acquired with experience on the job, specifically those related to cultural aspects.

There are very few opportunities identified by the experts regarding AI's influence on the provision of language services in Portugal. The expediency of the services is the most relevant one, together with AI's ability to provide prompt and more diversified vocabulary options/alternatives. It is worthy of note that Portuguese translators tend to value AI's creative features regarding the diversity of vocabulary it provides to enrich their texts. In contrast, 96% of Bulgarian translators agree that AI will never replace human creativity [18].

Concerning the overall scenario in Portugal, and according to the professionals' views, there are severe concerns regarding a significant reconfiguration of the career and loss of income and/or jobs, justified by automation and mainly by AI, which works at the operational and creative levels. In Bulgaria, the results depict AI, together with other digital technology, as becoming an "organiser of new activities within the profession", with translators becoming the system's teachers and not leaving the profession. Routine translations are automated and absorbed by AI, and new jobs are simultaneously created with the increase in the demand for editors of AI translated texts, as well as teachers of AI [18]. This is seen as a positive outcome of the penetration of AI in the career. However, Portuguese translators view the reconfiguration of the profession into reviewers or post-editors as a depreciation of the profession, leading to a reduction in income and volume of work. Thus, they do believe that AI will negatively impact the quality of work or employment conditions in their professional activities. Moreover, within the qualitative data collected, none of the professionals envision themselves having a tutoring or teaching role over AI, such as reported by [18], in the sense of perfecting it to improve it. What is

commonly recognized is the increased expediency of the provision of services, facilitated by AI and automation.

Among the perspectives of experts with a more positive attitude, some mitigation strategies for professionals are suggested:

1. *We have to adapt and keep up with the trends, otherwise many translators may not be able to maintain the income they are used to.* (P15, Pos. 1)
2. *I foresee a redefinition of the profession, the need for continuous training for professionals to adapt to other realities.* (P13, Pos. 1)
3. *It is up to us, both as professionals and consumers, to value our work, to call companies' attention to low-quality texts and content, and not let the market crush us, no matter how scary and exhausting the situation becomes.* (P07, Pos. 1)

The experts refer to the need of keeping up with the technological trends and invest in continuous training to help professionals incorporate AI as a competitive tool, allowing them to remain competitive and relevant. To this purpose, raising awareness regarding the poor quality of automated or AI-based translations among clients and companies is also believed to be necessary.

## 6. Conclusions

This study aimed to provide insights about the influence of AI on Portuguese LSPs' work performance and their perceptions about AI's impact in their careers.

The research shows that AI has critical effects on the provision of language services in Portugal.

The findings indicate that the respondents are generally familiarised (moderately to very familiar) with the use of AI (67%) but are unable to clearly identify (73.2%) if AI and/or automation technologies support tools that are commonly used within the profession. It was also possible to see that NMT is almost two times more frequently used than LLM, probably due to LLM platforms being more recent, but the overall frequency of use of AI technologies is between 25% and 50%.

The results also reveal that the professionals use AI mainly for reasons of speed, as a starting point for a translation, and for short texts. The reasons behind using LLM are similar to the ones indicated for NMT.

Overall, only 34% of the respondents consider these technologies as very or extremely useful, and 15.19% find them not useful at all. Still, it is interesting to note that the sense of usefulness is higher for those who were more familiar with AI, which means that once professionals get more acquainted with AI, they tend to see more value in it.

Regarding the influence of AI on their work performance, professionals believe that "Transcription", "Translation of general topics", "Terminology management", and "Technical/specific translations" are more prone to the influence of AI. Those with less positive effects would consist of "Interpretation" and "Subtitling", probably due to external variables that can influence the process.

Even though the answers were obtained from a balanced sample of experts regarding the positive or negative perspectives regarding AI's influence on the langue services market, when considering the effects on the Portuguese market, the negative outcomes emerge as more relevant. This happens because, even those who believe in a more positive influence also voice more general or more specific concerns with a non-positive perspective. The devaluation of experts and their career is at the top of these concerns and appears, sometimes, linked to the lack of formal regulation of the career, but not only.

In sum, one of the more voiced concerns was that experts may turn into mere reviewers and that the lowering of quality standards and increased speed of translation, will cause a severe reduction in prices. It is perceived that that machine translation may lead to a shift in the language market dynamics depreciating the perceived value of language services and the value of experts.

Given this context, we believe that the roles of human professionals may evolve to focus on more complex or specialized tasks, or depreciate, while AI tools will be used for

more routine or repetitive tasks. Many language tasks still require human judgment and context, such as creative writing, cultural translation, and interpretation, which are less amenable to some sort of automation.

Overall, AI has the potential to transform many aspects of life and create new opportunities and efficiencies. However, it is important to approach AI development and deployment responsibly and ethically to ensure that its benefits are maximised, and its potential risks are minimized.

This work is not without limitations. On the one hand, it is built on a convenience sample that, although relevant, may not be fully representative of the of the entire country. On the other hand, the research focuses only on the professional's view of the market, disregarding the perceptions of the clients they serve, which could be an added value to how the market is currently behaving and help determine future trends.

**Supplementary Materials:** The following supporting information can be downloaded at: https://www.mdpi.com/article/10.3390/informatics10040081/s1, File S1: Questionnaire.

**Author Contributions:** Conceptualization, C.T., L.O. and M.M.d.S.; formal analysis, L.O.; funding acquisition, M.M.d.S.; investigation, C.T., L.O. and P.D.; methodology, C.T. and L.O.; project administration, C.T. and L.O.; resources, C.T., P.D. and M.M.d.S.; supervision, C.T. and L.O.; validation, L.O.; visualization, L.O.; writing—original draft, C.T., L.O., P.D. and M.M.d.S.; writing—review and editing, C.T., L.O., P.D. and M.M.d.S. All authors have read and agreed to the published version of the manuscript.

**Funding:** This work is financed by Portuguese national funds through FCT—Fundação para a Ciência e Tecnologia, under project UIDB/05422/2020.

**Institutional Review Board Statement:** The study was conducted in accordance with the Declaration of Helsinki and approved by the Dean of the Institution. Ethical review and approval were waived for this study because any unknown and unwilling sensitive data about the participants was not collected. Moreover, the data voluntarily provided by the participants is stored in institutional Limesurvey servers, and participants were informed that they can exercise their right of access, rectification, cancellation, or opposition at any time. Finally, the data were statistically treated in a consolidated manner.

**Informed Consent Statement:** Informed consent was obtained from all subjects involved in the study.

**Data Availability Statement:** Data are contained within the article.

**Conflicts of Interest:** The authors declare no conflict of interest.

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
