# Peer review of "Artificial Intelligence: A Blessing or a Threat for Language Service Providers in Portugal"

_informatics, doi:10.3390/informatics10040081_

Round 1
Reviewer 1 Report
This paper aims to survey on AI and Automation technology in the perspective of translation. It is worth to do it. It concentrates on Portugal LSPs (translation service)
There are several concerns as follows
1) From table 5, author score 1-9, however it is not clear on the definition of score. I can catch 0 and 1 based on the explanation, but not 1-9
2) From Table 6, the author try to compare the frequency of use of AI technologies on the job, based on three technologies. It is quite difficult to understand why three technologies are raised. It seems to me that it is overlapping among three of them. Rational on using them should be clarified.
3) There are a lot of opinion from real users in a lot of perspectives. However, it lacks of the analysis based on the survey and the recommendation. IT should be better to state clearly and link to the objectives of this work.
4) It is better to understand the rational of translation behind in ChatGPT. It will help the comparison become clearer.
English is quite easy to understand, however, the flow of paper should be edited.
Author Response
Thank you for revising our work and for the kind comments and recommendations, which helped us to improve it. Please find attached the answers to the recommendations.

Reviewer 2 Report
Overall
The authors have identified a novel niche, specifically the impact of LLMs on MT and in turn the jobs of professional interpreters and translators. The paper is well-written but lacks some details that are needed to make an evaluation of the results. The argument appears logical. The method is not fully described and sufficient details are not provided for the statistical tests. I have listed a number of issues for the authors to respond to. Most should be straightforward to address.
Major issues
1. Abstract
There is a jump from research on the impact of LLMs on jobs in the US to Portuguese LSPs. In fact, almost all the LSPs surveyed are based in Portugal. There is therefore an assumption that AI will impact other countries, too (which I believe is true).
2. Introduction
Line 40 Google Translator -- > Google Translate
3. Theoretical background
Line 140 LML --> LLM
4. Method
The sample size of the questionnaire survey is rather small, but that is not my criticism. What is more important is how representative is this sample of the population of Portuguese LSPs? Please provide some data on the (estimated) number of Portuguese LSPs that it is possible to better evaluate whether any results are statistically generalizable.
5. Method – Questionnaire
Mention the number of core questions and the number of contingent questions.
Please provide an English translation of the questions on the questionnaire in an Appendix
6. Method – Analysis
The methods of analysis of the data should also be included here rather than built into the results.
7. Results
Lines 218-21 Please provide more details for parametric statistical tests here. You did not state what you were comparing (degree to non-degree?)
8. Table 2
The rows seem to have a problem.
9. Lines 244-255 and others
Please share the statistics whenever you conduct statistical tests.
10. Figures –bars
The numbers in the coloured bar are difficult to read. Ensure that you use light numbers on dark backgrounds and dark numbers on light backgrounds.
11. Quiz
Line 330 – Should this quiz not be mentioned in the methods? What did it involve?
12. Figure 7
Increase the font size for the labels in the Codes map.
13. Tables – verbatim quotes
In qualitative research reporting verbatim quotes help establish depth. However, presenting quotes in a table format seems rather odd. I suggest the authors simply use a numbered list with subheadings if needed.
14. Length – verbatim quotes
I understand the authors may want to provide more context, but some quotes in my view are unnecessarily long.
15. Table 13 – Points/Commas
Please use the English version e.g. 12.34 rather than 12,34 in all tables
16. References
The reference list is rather thin, but in its defence, most sources are recent with a number published in 2023.
Author Response

(The authors gave the same response as above.)

Round 2
Reviewer 2 Report
Overall
The authors have tried to address most of the issues raised in my initial review. There are, however, still some outstanding items to deal with.
Outstanding issues
1. Abstract
Believing that this context will spread worldwide, this article, following a mixed-methods survey-based research, provides insights into the Portuguese Language Service Providers (LSPs) awareness and knowledge about AI, specifically regarding neural machine translation (NMT) and large language models (LLM), its actual use and usefulness, as well as their potential influence on work performance and labour market.
The original version is better. “Believing that this context will spread” does not make sense.
2. Table 2
The left-hand columns do not relate to the right-hand columns. This single table should be spilt into a more readable format. The items on a single row should relate to each other but clearly male, 17, 21.5% and Freelance Translator do not. This table is six separate tables merged into one. There are many ways to reformat this, but the most important point is not to place items on the same row that are unrelated. Personally, I would separate it into two tables. Table 2a would be four columns and would contain Gender, Country, Education. Table 2b would have multiple columns and could contain the remaining items. Feel free to adopt another approach but make sure it is reader-friendly.
3. Table – verbatim quotes
In qualitative research reporting verbatim quotes help establish depth. However, presenting quotes in a table format seems rather odd. I suggest the authors simply use a numbered list with subheadings if needed.
I understand the necessity to assign a code (APA 7 and standard practice), but I see no reason to use a one-column column table to display verbatim quotes. In fact, the use of the table is distracting and disrupts the flow. I suggest the others follow this approach with or without using a numbered list.
1. “verbatim quote” (unique identifier 1)
2. “verbatim quote” (unique identifier 2)
3, “verbatim quote” (unique identifier 3)
none
Author Response
Thank you for you kind comments and suggestions. Please find our answers in the document attached.
